# IoT Adoption and Application for Smart Healthcare: A Systematic Review

**DOI:** 10.3390/s22145377

**Published:** 2022-07-19

**Authors:** Manal Al-rawashdeh, Pantea Keikhosrokiani, Bahari Belaton, Moatsum Alawida, Abdalwhab Zwiri

**Affiliations:** 1School of Computer Sciences, Universiti Sains Malaysia, Penang 11800, Malaysia; bahari@usm.my (B.B.); moatsum.alawida@adu.ac.ae (M.A.); 2Department of Computer Sciences, Abu Dhabi University, Abu Dhabi 59911, United Arab Emirates; 3School of Dental Sciences, Health Campus, Universiti Sains Malaysia, Kelantan 16150, Malaysia; abdalwhab@student.usm.my

**Keywords:** IoT, IoMT, IoT adoption, systematic review, adoption theories, adoption factors, Internet of Things in healthcare, machine learning (ML), deep learning (DL)

## Abstract

In general, the adoption of IoT applications among end users in healthcare is very low. Healthcare professionals present major challenges to the successful implementation of IoT for providing healthcare services. Many studies have offered important insights into IoT adoption in healthcare. Nevertheless, there is still a need to thoroughly review the effective factors of IoT adoption in a systematic manner. The purpose of this study is to accumulate existing knowledge about the factors that influence medical professionals to adopt IoT applications in the healthcare sector. This study reviews, compiles, analyzes, and systematically synthesizes the relevant data. This review employs both automatic and manual search methods to collect relevant studies from 2015 to 2021. A systematic search of the articles was carried out on nine major scientific databases: Google Scholar, Science Direct, Emerald, Wiley, PubMed, Springer, MDPI, IEEE, and Scopus. A total of 22 articles were selected as per the inclusion criteria. The findings show that TAM, TPB, TRA, and UTAUT theories are the most widely used adoption theories in these studies. Furthermore, the main perceived adoption factors of IoT applications in healthcare at the individual level are: social influence, attitude, and personal inattentiveness. The IoT adoption factors at the technology level are perceived usefulness, perceived ease of use, performance expectancy, and effort expectations. In addition, the main factor at the security level is perceived privacy risk. Furthermore, at the health level, the main factors are perceived severity and perceived health risk, respectively. Moreover, financial cost, and facilitating conditions are considered as the main factors at the environmental level. Physicians, patients, and health workers were among the participants who were involved in the included publications. Various types of IoT applications in existing studies are as follows: a wearable device, monitoring devices, rehabilitation devices, telehealth, behavior modification, smart city, and smart home. Most of the studies about IoT adoption were conducted in France and Pakistan in the year 2020. This systematic review identifies the essential factors that enable an understanding of the barriers and possibilities for healthcare providers to implement IoT applications. Finally, the expected influence of COVID-19 on IoT adoption in healthcare was evaluated in this study.

## 1. Introduction

With increasing chronic illnesses, the number of patients in dire need of medical intervention is rising rapidly. This has invariably put pressure on healthcare services and delivery. Healthcare administrators, physicians, nurses, and other health professionals are facing increasing pressure to respond to the growing demands of both the public and the private sector on health-related matters. Additionally, the increasing cost of medical care has imposed a significant effect on the quality of people’s lives. The development of healthcare systems requires a concerted effort to seamlessly integrate with the Internet of Things (IoT), especially for ameliorating day-to-day challenges arising in the sector. Recent developments in the health sector have consistently shown that combined technologies have the potential to improve healthcare services and assist healthcare professionals in the optimal and efficient delivery of healthcare solutions [1,2,3]. IoT is a new paradigm in technology which provides a conglomerate of novel services for the next wave of technological innovations [4]. IoT enables things (such as devices, cars, houses, people, and animals) to communicate with one another and with users over the internet network, thereby becoming an integral part of the Internet [5,6]. Furthermore, cloud computing services are used in IoT applications to create correct composite services by composing existing atomic services for IoT service-based applications [7]. IoT applications also give users a lot of advantages, such as the ability to choose the best opportunity in each situation, to make decisions, to manage resources, and keep an eye on the environment’s cloud resources [8]. A big part of the IoT is RFID, sensor technology, nanotechnology, and embedded intelligence technology. Each of the aforementioned technologies are being used to advance IoT applications for various purposes [2,9]. One of the major targets of the health sector is to realize high-quality healthcare delivery with low cost; the IoT has the power to make this real. For instance, the incorporation of sensor systems helps with better patient monitoring, leading to fewer tests, fewer unnecessary appointments and, consequently, lower costs. Hence, IoT technology is a key player in the early diagnosis and early intervention of diseases [10].

Significantly, the Internet of Medical Things (IoMTs) [11] or IoT (for the purposes of this paper, the terms IoT in healthcare and IoMT will be used interchangeably) will support the digital revolution, particularly in healthcare products. The developed IoMT applications prototype such as wearable devices enable patients, elderly people, or people with chronic diseases to remotely monitor their health status. In such cases, IoT applications can help in an emergency to quickly warn and alert caregivers or physicians of the elderly person [12].

Even though IoMT and its supporting technologies have been proven to mitigate health problems, such as medical errors, failure, ineffective workflows, and all evident benefits of IoT technologies in the healthcare sector, IoT systems are not fully integrated into healthcare organizations yet [13]. Additionally, IoT developments in the health sector have remained slow in terms of its implementation and adoption in other industries [14].

In light of the low adoption of IoT in the healthcare sector, it is difficult to implement it if users are not ready. Moreover, the decision to adopt IoT application requires a structured approach that is capable of identifying the technological and operational structures. In- deed, technology adoption is one of the mature areas of research in information systems, especially IoT adoption [15]. Huge investments are being made by companies and governments to adopt innovations that have the potential of bringing a paradigm shift in the user’s lifestyle; for instance, the IoT technologies [16]. Several studies expect that healthcare professionals will have new responsibilities through using IoT [17]. Although the IoT can provide an improved and better approach to healthcare management, its end-user adoption is still very low especially among health-care professional staff [15].

Therefore, the objective of this work is to analyze and statistically classify the present research on the adoption of IoT among professional staff in healthcare, and to gain a comprehensive understanding of the adoption of IoT processes in healthcare. To meet the aforementioned objectives, several research questions (RQs) were formed. The novelty of this systematic literature review (SLR) provides the most recurrent adoption theories for IoT and the most recurrent factors that have a significant effect on its adoption in healthcare. This study classified IoT adoption factors into five categories which are related to individual factors, technology factors, security factors, health factors, and environmental factors. This paper provides an overview of the articles that are related to the adoption of IoT applications in healthcare from 2015 to 2021. To the best of our knowledge, there is no literature covering these points yet [18,19,20,21,22].

This study is more distinguished and unique as it explores new areas that previous studies rarely explore, or may have been referred to briefly. Furthermore, this study reviews the effective factors and criteria influencing IoT healthcare systems adoption. It used questionnaires, interviews, and expert opinions on the effective use of IoT. It also reviewed prior related research that provided new conceptual models related to the intention to use of IoT as a new technology. This SLR is constructed as follows: Section 2 provides an overview of IoT in healthcare. Section 3 elucidates the advantages of IoT in healthcare systems. The review strategy is provided in Section 4. Section 5 gives details on the characteristics of the included studies. Section 6 reviews IoT for testing and tracing. Section 7 discusses wearable devices used for IoT applications in healthcare. Section 8 includes regulations and procedures for IoT during pandemics. Section 9 discusses the findings of this SLR, which contains gaps and implications for future research, limitations, challenges of effective IoT’s adoption, research directions, pervasive challenges across all verticals, healthcare during COVID-19 pandemic challenges, and data protection and privacy. Finally, Section 10 wraps up the paper with concluding remarks.

## 2. Overview of Deep Learning for IoT in Healthcare

IoT systems are made up of a large number of heterogeneous devices that are scattered across the network and generate a constant stream of data [23]. To design successful IoT applications, we often follow a workflow model composed of five components: formulation of the question, data gathering, data processing, visualization, and assessment. Even though obtaining hidden information and inferences from IoT data is a promising way to improve our lives, it is a difficult task that can not be done with traditional paradigms [24]. Indeed, with the birth of IoT devices, artificial intelligence (AI) has been introduced that uses continuous monitoring to help in illness diagnosis, notifying caregivers or physicians through an alert system. Apart from that, these gadgets may also be used to assist in decision-making through a decision support system (DSS). A significant advantage of this change was the transition of duties from a manual, stressful, and time-consuming process to a more intelligent, automated, and time-efficient one. Moreover, there have been occasions when medical practitioners have been unable to care for patients owing to a lack of information about emergency situations, resulting in disastrous choices, if not death. These machines are educated using specialized artificial intelligence algorithms, most often referred to as machine learning (ML) and deep learning (DL) algorithms [25]. DL has been actively utilized in many IoT applications in recent years [26]. It is a subset of machine learning (ML), and it is computationally intensive and costly. One of the issues is integrating DL techniques with IoT in order to increase the overall efficiency of IoT applications. Combining these strategies while maintaining a balance of computing cost and efficiency is critical for the next-generation of IoT networks [24].

Deep learning (DL) will be critical in developing a smarter IoT because it has demonstrated remarkable results in a variety of fields, including image recognition, information retrieval, speech recognition, natural language processing, indoor localization, and physiological and psychological state detection, amongst others. More also, these services serve as the foundation for IoT applications. Indeed DL has been actively utilized in many IoT applications in recent years [26,27].

## 3. Advantages of IoT in Healthcare Systems

The following are the primary benefits or advantages of IoT technologies in healthcare systems that impact their adoption:Cost savings: by being able to meet and assess patients remotely, the cost of in-person visits can be lowered [28,29,30]. Furthermore, with the introduction of home care devices, many patients can now be hospitalized and monitored at home.Treatment outcomes: because the monitoring is consistent, continuous, and automated, all data is kept in the cloud and provided to the doctor on a regular basis; the treatment processes were carried out correctly. The adoption of this strategy can ensure that medical care is provided as soon as possible to examine the recovery process [31].Disease management: by consistently recording and reporting a person’s health indicators, diseases can be discovered and treated before they progress [32].Error reduction: detailed and precise data collected automatically and free of human error can significantly reduce the rate of medical errors and their associated financial and critical costs [33,34].Patient satisfaction: some factors such as the emphasis on the patient’s requirements, data accuracy, timely treatment, cost reduction, reduction of repeated visits, recording of the recovery process, and, most importantly, the patient’s active participation in the treatment process, have a positive impact on the patient [35].Medication management: IoT assists patients in the precise use of drugs, as well as helping pharmacies and healthcare facilities in preventing drug waste [35,36].

## 4. Review Method

In this study, the SLR method was used, which is a way to find, analyze, and interpret all of the papers that have been written about a certain topic or research question [37]. The SLR process includes recognizing research, research questions, search strategy, study screening process and methods, quality evaluation, data extraction technique, and extracted data synthesis [37]. This systematic review was conducted for capturing relevant literature from different sources, focusing on the following objectives:To explore conceptual frameworks for the adoption of IoT in healthcare.To illustrate the future adoption of IoT in healthcare.Overall, any SLR should be able to synthesize and analyze existing data on any subject, look for research gaps, and suggest the future direction on that subject [37]. Through investigating these objectives in detail, this review will make a significant enrichment in understanding the future adoption of IoT applications in the healthcare domain.

### 4.1. Information Sources

For this SLR, the articles were searched for in the following nine digital databases: Google Scholar, Science Direct, Emerald, Wiley, PubMed, IEEE, MDPI, Springer, and Scopus as represented in Figure 1.

We carried out a search of the literature published between 2015 and 2021 related to IoT adoption among professional staff and IoT adoption in healthcare. Moreover, studies were selected from the databases using the following keyword combinations: Adoption, IoT, Nurses, (Adoption and IoMT and Nurses), (Adoption and IoT and Nursing care), (Adoption and IoT and Physician), (Adoption and IoMT and Physician), (Adoption and IoT in healthcare), and (Adoption and IoMT in healthcare).

### 4.2. Selection Criteria

The inclusion criteria for included review papers are: 

1-Published between 2015 and 2021.

2-Written in English.

3-Available as full text.

4-Published in a peer-reviewed journal.

5- Articles and conference papers.

6-Articles investigating the adoption of IoT in healthcare.

The exclusion criteria are: 

1-Duplicate studies.

2-Not written in English.

3-Not related in the adoption of IoT in healthcare.

4-Not available as full text.

5-Outside the time frame.

### 4.3. Research Question

Specifically, the goal of this part is to define the study questions in order to better understand the difficulties associated with the adoption of IoT in healthcare. The identification of the research question aids in the definition of the scope and goals of this study. The aims of this research are to investigate the area of IoT adoption in the healthcare sector and how the results of this study can be used to create a vision of the factors that assist the health industry in using IoT technology. To meet the research objective, the below four research questions (RQ) are formulated:RQ1:What are the primary adoption areas that were selected by studies (adoption of IoT devices or end users’ adoption for the chosen studies)?RQ2:What theories/models were used in the studies?RQ3:What constructs are being employed in the studies?RQ4:Which kind of techniques are employed for data analysis in the chosen studies?RQ5:What are the research gaps on in the current studies which are related to the use of IoT in the healthcare sector?

### 4.4. Quality Assessment

The quality assessment (QA) is defined as an accurate assessment about the general quality of the selected papers (N = 22). Several initiatives aimed at improving the overall quality of the search process were carried out throughout the assessment. Thus, to avoid any potential effects from previous searches, the online searches were conducted in incognito mode. The authors manually extracted relevant papers and articles from the first searches they conducted. In addition, when the abstracts were reviewed in detail, it was determined that certain publications had to be kept in the investigation and others needed to be dropped. In addition, a quality process for this SLR is developed by formulating five QA criteria as presented below:Q1:Is the paper’s topic related to the adoption of IoT technology and its applications?Q2:Does the paper use adoption theories?Q3:Does the paper have theoretical framework and constructs?Q4:Does the paper explicitly present the research methodology?Q5:Is the procedure for collecting the research data clearly outlined in the paper?

A QA score is used to evaluate the study’s overall quality as shown in Table 1. The study’s overall quality is ranked on one of three quality levels: “high”, “medium”, and “poor” [38]. Research that entirely fulfills a quality requirement is given a score of 1 based on the resultant load score at the start of the investigation. Second, a score of 0.5 is provided to a candidate that just partly meets the requirements for consideration. After that, each item that obtains a score of 0 is awarded a score of 0 since it does not fulfill a quality standard, as defined by the five criteria for evaluation. With regard to the criteria, the greatest possible loaded score for each research is 5, and the lowest possible loaded score is 0 for each study. (see Table 1). This article summarizes the findings of the QA process that was used on the 22 selected studies. The results show that 18 studies (82%) were classified as “high-quality”, whereas 4 (18%) were classified as “medium-quality”. There were no studies classified as “low-quality”. As a consequence of this decision, no more research was omitted from the total. As a consequence, this SLR is mostly made up of 22 original papers that were carefully chosen.

### 4.5. Results

Here, the prime concern was to find out the effective factors for the adoption of IoT in healthcare. The effective factors are those factors that affect the use of the healthcare system and also have an impact on user satisfaction [39]. Moreover, it also has to do with investigating the adoption of the IoT conceptual framework in healthcare, and adoption theories that are used in the adoption of IoT in the healthcare domain. The criteria for inclusion have been established as papers using the search keywords, which are mainly adoption, IoT in healthcare, and adoption of IoT by nurses, physicians, and patients.

Besides the title, abstract, and keywords of the papers that have been found, a list of notable articles has been made based on certain criteria. On the other hand, studies that focus on technical concerns and security challenges were left out of the analysis. The abstracts and titles of some of the studies were reviewed separately, to see if they met the criteria for being included. The procedure for search and selection of research material is illustrated in Figure 2.

#### 4.5.1. Data Extraction and Organization

Based on the initial author’s name and year of publication, as well as the kind of participants, the study design, and the geographical location of the research, if there is a theoretical framework, data were retrieved.

Table 2 shows the results of the quality evaluation for each of the selected studies and the elements extracted from each study. The authors extracted the data and double-checked it before publishing it. During the search approach, we reviewed all of the titles and abstracts of the selected papers. After that, we each looked at the full text of the articles that had been pre-selected and agreed on which one to choose. In addition, we did data extraction independently using a verified data extraction grid that was built based on the study criteria that we used. This generic data extraction grid has been designed and tested to categorize published publications according to existing theoretical frameworks, adoption theories, constructs, study locations, and design approaches. The data extraction grid in the Microsoft Excel program was filled in manually.

#### 4.5.2. Included Studies

In the beginning, this research search strategy provided a total of 17,597 papers from databases such as PubMed, Wiley, Google Scholar, SCOPUS, Emerald, and Science Direct. The remaining 5327 papers were further screened after eliminating 1227 papers in the detection phase (unrelated to healthcare, technical papers, security issues, literature reviews, and duplicate studies). A total of 75 studies were considered worthy, but due to not meeting the inclusion criteria, 53 studies were excluded. Eventually, based on the research goals, inclusion and exclusion requirements, 22 studies were included in this study and the full text of all the included studies has been retrieved.

## 5. Characteristics of Included Studies

In this review, we started conducting a search among publications between 2015 and 2021. However, we observed that in 2015 there were no publications related to our scope of study in all databases that we used. Figure 3 illustrates the studies’ type. Few studies did not use theoretical frameworks. Figure 4 illustrates the number of articles growing exponentially from 2016 to 2021 [20,40]. In general, there are several theories, models, and frameworks that have been developed to explain user adoption of new technologies. The models introduce factors that can affect user acceptance such as the technology acceptance model (TAM), theory of planned behaviour (TPB), diffusion of innovation theory (DIT), theory of reasoned action (TRA), unified theory of acceptance and use of technology (UTAUT), unified theory of acceptance and use of technology2 (UTAUT2), and Seddel model [41] (see Table 3).

Indeed, some of the selected studies for this SLR used other theories besides adoption theories to understand the effective factors in the adoption of IoT in healthcare, such as theories related to security and healthcare. For instance, cybernetic control theory (CCT), which is based on the idea of obtaining timely feedback, breaking down the deviations from the expectations, and taking important choices to address the deviations [42]. The theory of protection motivation (PMT) is concerned with the cognitive processes that mediate changes in attitudes and behaviors in healthcare [10]. Finally, the health belief model (HBM) is used “to describe and foresee health behaviors, to know the relationship of health behaviors, and utilize health services and practices systematically” [43].


sensors-22-05377-t003_Table 3Table 3The theories and their constructs that were used in each study.StudyAdoption’s TheoryConstructs [44](TRA)(TPB)(TAM)Adoption intentionPerceived behavioral controlPerceived usefulnessPerceived ease of use
Subjective norm [40](UTAUT)Performance Expectancy
Effort expectancy
Social InfluencePerceived
RiskFacilitating
ConditionsFinancial Cost
Behavioral Intention [45](UTAUT)
(UTAUT2)Performance Expectancy
Effort expectancy
Social Influence
Facilitating Conditions
Prereceived creditability [46](UTAUT)Performance Expectancy
Effort Expectancy
Social Influence
Perceived Risk
Facilitating Conditions
Perceived trust
Behavioral Intention
Age, gender, experience [47](BRT)Ubiquitous Reflective
Reasons for Convenience Reflective
Ubiquitous Reflective
Relative advantage reflective
Compatibility Reflective
Reasons against Usage barrier reflective
Risk barrier reflective
Traditional barrier
Reflective Attitude reflective
Adoption intention reflective
Value of openness to change [10](TAM)
(IDT)
(PMT)Perceived Advantage
Technological Innovativeness
Compatibility Trialability
Image
Perceived Vulnerability
Perceived Severity
Perceived Privacy Risk
Cost
Perceived Ease of Use
Attitude [48](DOI)
(TAM)Perceived usefulness
Usefulness
Necessariness Improvement
Perceived usefulness [49](UTAUT)Performance Expectancy
Effort expectation
Behavior intention
Behavioral to use [14](DOI)Perceived usefulness
Perceived easy to use
Computer self-efficiency
Personal innovativeness
Computer anxiety
Services quality
information quality [50](FAHP)Economic Prosperity
Environmental Protection
Quality of Life [51](TAM)
(TPB)
(TRA)
(UTAUT 2)Trust organization
Trust Provider trust treatment
Trust technology [52](TAM)
(TPB)
(TRA)
(SE Theory)Interpersonal influence, self-efficacy
Attitude toward a wearable device
Health interest, Perceived value trustworthiness [53](IDT)
(TAM)Perceived ease of use
Behavioral intention cost
Trialability compatibility attitude
Privacy, Image
self-efficiency
Perceived usefulness [54](UTAUT)Not Mentioned [42](CCT)Information Pervasiveness
Care Process Improvement
M-IoT Adoption Care
Service Efficiency [55](TAM)
(UTAUT)Performance expectancy [43](HBM)
(UTAUT)Effort Expectancy
Social Influence will
Facilitating condition
Performance expectancy
Perceived severity
Use behavior Trust
Doctor’s patient relation [56](TAM)Security
Privacy, Trust in IoT
Risk perception Familiarity
Attitude [57](TAM)Behavioral Intention to Use
Perceived Usefulness
Perceived Ease of Use
Attitude
Perceived Connectedness
Perceived Cost Privacy
Concerns Perceived Convenience [58]Not MentionedCritical data management
Unreliable results accuracy, security
Unreliable results accuracy
Unaffordable technology for low-income groups
lack of clear regulations.
Critical data management
Lack of clear regulations. [59]Saddon ModelPersonal innovativeness
E_loyalty
Usefulness
Personal innovativeness [60](TAM)
(HBM)Perceived usefulness
Consumer innovativeness
Health information accuracy
Reference group influence
Health beliefs
Privacy Protection


As Figure 5 depicts, the majority of studies used the TAM, UTAUT, and its updated 296 version UTATUT2 adoption theories [55]. However, TPB, TRA, DOI, BR theories [44], the IDT theory, HBM theory, Seddon model, CCT theory, IDM theory, PMI theory are used in fewer articles [42,44]. The studies about IoT adoption in healthcare were conducted in various countries including France, Spain, Germany, Sweden, Turkey, Hong Kong, and Israel. Developed and developing countries, such as India, Pakistan, Malaysia, Saudi Arabia, Iraq, Oman, and the Kingdom of Saudi Arabia from Asia and Latin America also recorded a large body of research in IoT adoption [43,53,57].

The majority of studies used quantitative research design (17.77%) [43,46]. One study employed a qualitative design through the use of focus groups (1.4%) [55], and representations of data collection methods. A mixed-method design was used in one study (1.4%) [46,54]. The rest of the studies did not mention the design methods they used [50]. The types of IoT applications studied were the following: a wearable device [55,61], general IoT devices (without specific any type) (9 papers), monitoring devices (2 papers) [52,58], rehabilitation device (1 paper), telehealth and behavior modification (1 paper) [42], smart city, and smart home (1 paper). Clinicians, nurses, medical workers, pharmacists, and other healthcare practitioners were involved in the included research (such as nutritionists, social workers, occupational therapists, and care services). Some other studies (4 papers) exclusively involved physicians [43,45,50,52]. The rest of the publications focused on patient (4 papers) [40,51,57,58], and end-users of IoT device in healthcare (9 papers) [40,52] (see Table 4).

### 5.1. Overview of IoT Adoption Factors

In the results, 90 elements were identified as barriers or facilitators for IoT technology adoption and were classified in the different categories of factors from the extraction grid. As Figure 6 shows, these elements were classified as facilitators for IoT adoption in healthcare and barriers.

Indeed, a total of 61 elements (67%) are concerned with the category factors related to the IoT adoption in healthcare characteristics, in which 41 elements of them were identified as barriers and 20 elements as facilitators. The most repeated adoption factor was perceived usefulness [40,48,53]. Perceived usefulness is defined as an individual’s perception that the utilization of IoT device will improve their performance of daily activities [53]. Perceived ease of use was another frequently mentioned factor [10]. Perceived ease of use is defined as an individual’s perception that the utilization of IoT devices will be effortless [10,59].

As a result, it was critical for professionals to perceive the usefulness and ease of use of technology in their workplace; otherwise, there would be less incentive to use them. Other factors related to IoT in healthcare characteristics are cost issues, privacy, security concerns, and healthcare issues. The cost issues, privacy, security concerns, and healthcare were seen as barriers to the adoption of IoT in healthcare. Indeed, specialists were worried about the safety and confidentiality of the data contained in, and transmitted by, these technologies, as well as the possibility of device theft. Additionally, the cost of the IoT technology and its applications were perceived as barriers to IoT adoption in healthcare, and healthcare issues. The specialists are worried about the health risk if the technology is used in their health activities. Other factors identified in this category were individual factors, technology factors, security factors, health factors, and environmental factors.

### 5.2. Individual Factors

Individual factors in IoT adoption represented 16 elements (26%) among the total extracted elements. As shown in Figure 7 the most common factor identified was social influence factor, which is repeated six times [40,43,45,49,60]. Generally, professionals thought that IoT technology adoption is affected by social impact and conviction to receive support to use IoT technology in their activities, and it is used as facilitators more often than as barriers [40,45,49].

The next factor for individual adoption is attitude. It was mentioned five times [10,44,53,56,57]. Personal inattentiveness was mentioned four times as a facilitator [10,14,59,60], and self-efficacy, age, gender, and experience were mentioned three times each. The aforementioned factors are mentioned as facilitators more than as barriers [14,45,52]. Professionals believed that IoT adoption in healthcare could improve these factors. Finally, the next sets of factors are adoption intention, compatibility, and image. They were mentioned two times [10,52]. They were seen as barriers more than as facilitators. Furthermore, anxiety, interpersonal influence, value of openness to change, user satisfaction, e-loyalty, and perceived contentedness were mentioned only one time. They were perceived as barriers [59]. Professionals believed that IoT adoption in healthcare could slow down due to these barriers.

### 5.3. Technology Factors

IoT adoption technology factors represented 19 elements (31%) of the extracted elements. As shown in Figure 8 the most common factor identified was perceived usefulness factor, which was repeated six times [14,46,48,53,59,60]. As mentioned before, it was regarded as a facilitator. The professionals believed that the perceived usefulness factor in IoT adoption in healthcare is an essential factor as it improves their working conditions. The next factor related to IoT adoption in healthcare is perceived ease of use. It was repeated five times as a facilitator. The specialist is looking for technology that is easy to use during their duties. Thus, it is believed to be an important factor [14,48,53,57,59]. Performance expectancy factor is repeated four times [43,45,46,48], and effort expectation factor is repeated three times [14,43,46]. Both factors appeared as facilitators. Effort expectations are streamlined with the consensus based on the belief of the professionals that the use of IoT technology would be without effort. Performance expectancy is based on the belief that the use of the IoT technology would improve their performance [46], and from the professionals’ perspective, these factors are considered as strong motivation to adopt IoT technology in their work. Additionally, trialability, and reasons for convenience reflective factors were repeated two times, [10,47]. Moreover, information quality, perceived advantage, technological innovativeness, perceived creditability, compatibility reflective, relative advantage reflec- tive, ubiquitous reflective, unreliable results accuracy, information pervasiveness, features deemed most useful, utilitarian hedonic benefits sought, features deemed most useful, and critical data management factors were repeated one time [10,47,55], and they mention it as barriers.

### 5.4. Security Factors

The main factors related to security for IoT adoption represented seven elements (11%) of the extracted elements. Perceived privacy risk, as shown in Figure 9 is the most common factor identified in the selected papers. It was repeated seven times, followed by trust which was repeated six times. In each instance, they were considered as facilitators [43,46,47,53,56,57,60]. Furthermore, perceived vulnerability, trust organization, trust provider, trust treatment, and trust technology are considered as barriers. They were repeated once each [51]. Indeed, the professionals believed that the security issue is a critical issue, especially in the case of using IoT technology in healthcare. This is because healthcare information needs a high level of privacy, and other professionals believe they could be threatened, misused or misinterpreted while using IoT technology.

### 5.5. Health Factors

Health-related factors are essential to these studies. As shown in Figure 10, the most common factors identified in this category respectively are perceived severity and perceived health risk, where their repetition is two times [43,60]. Perceived severity “is explained as the degree of harm from unhealthy behavior” [43]. Furthermore, perceived health risk is defined as the degree to which the professional believes that using IoT will bring health problems [62]. This is followed by health interests, necessaries, health information accuracy, health beliefs, care process improvement, and care service efficiency, respectively; these appeared once each [52,55]. In this category, the studies consider these factors to be important, as they attempt to understand the adoption of IoT in health conditions. The professionals believed that using IoT technology in their duties could cause health problems related to the patient during the care process.

### 5.6. Environment Factors

The last category encompasses external factors that exist inside the organizational environment and accounts for 7 elements (11%) of the extracted elements. Financial cost factor, as shown in Figure 11, is the most common factor, repeated four times [10,46,53,57]. The next facilitating conditions factor was repeated three times [43,45,46]. Other factors such as environmental protection, quality of life, traditional barrier reflectivity, lack of clear regulations, and technical infrastructure were spread equally at once each [47,50]. All environment factors are considered as barriers. The professionals believe that these environmental factors, especially both the cost factor and the facilitating conditions factor, affect the adoption of this technology. The higher the cost, the more the negative impact increased, particularly in long-term costs of the technology. The costs of the system and applications were mentioned in the studies. Moreover, if facilitating conditions are not applied to this development, the more negative impacts will be present in its adoption.

## 6. IoT for Testing and Tracing

COVID-19 testing and tracing enabled by the IoT can cut down the spread of the disease, which is very important in the fight against it. People are now more likely to use IoT to test and trace things, which is accelerating IoT adoption [63,64,65]. Thus, Figure 12 shows a test and trace system. Such advanced test and trace systems are composed primarily of two layers: data acquisition and data integration.

Data is gathered. Mobility data comes from a variety of sources, such as an immigration database, Global System for Mobile Communications (GSM)- and Global Positioning System (GPS)- enabled mobile phones, and Quick Response (QR) codes that can be tracked. This helps Taiwan keep track of both foreign and domestic travelers. Moreover, smart city resources are used, such as CCTV cameras to keep an eye on things [66]. Additionally, credit card transactions are monitored and recorded in order to discover consumers and forecast their behavior [67]. Individuals’ hospitalizations are also logged and monitored to assist in locating missing persons [68].

A large number of countries and regions, including the United Kingdom, South Korea, Germany, Spain, Vietnam, and Taiwan, have adopted digital test-and-trace efforts [65,69]. While some of these attempts were unsuccessful in tracking the spread of COVID-19, nations that heavily used technology in their solutions did far better in their battle against this pandemic. Ref. [70] reported that the United Kingdom’s test and track system, which used around 27,000 contact tracers, had been unable to reach 21% of persons who tested positive for the week of 2–8 July 2020. Additionally, although 79% of those contacted identified 13,807 close connections, only 71% were contacted and urged to self-isolate.

Taiwan, on the other hand, had improved protocols in place (as a result of lessons learned from the 2003 SARS pandemic) and was able to suppress the initial COVID-19 wave. That is, as a result of swift and effective policy choices and widespread use of digital technology [71,72]. This indicates that a sophisticated technology-based test-and-trace system may be a viable tool for pandemic preparedness, provided all other safety measures are followed and rigorous policy choices are enforced. Figure 12 illustrates a structure for testing and trace system solutions.

A data integration layer combines and delivers data from disparate sources to the proper departments [73]. For example, the Centers for Disease Control and Prevention (CDCP) has access to the Immigration and National Health Insurance (NHI) databases [71]. Local governments are given access to location data. For example, surveillance data from a smart city setup is shared with police officers in order for them to take appropriate action [71]. Data extracted from various sources, such as public transportation and shopping malls, are also shared with the general public to assist them in making informed decisions about their daily routines [74].

## 7. Wearable Devices

While wearables such as smart watches, smart bands, and finger rings have long been available, the spread of COVID-19 has led to an upsurge in their appeal. When it comes to fighting COVID-19 and other future pandemics, wearable technology has a lot of promise [63,75,76]. Wearable technologies may also be used to broadcast health information [77]. Monitoring and connection tracing capabilities enforce social separation [78], provide tracking and contact tracing capabilities [79], enforce social separation [80], and provide mental healthcare [81] by continuously measuring an individual’s cognition and mood, allowing individualized therapy interventions [82].

As a consequence of these and other comparable uses, wearable devices are becoming more popular. Smart wearables, according to Papa et al. (2020) [82], have the potential to transform healthcare. By 2024, the wearable industry is expected to reach USD 64 billion, according to Global Data [83,84]. A short discussion of recent successes in the battle against COVID-19 will follow.

A COVID-19 identification method has been created by WHOOP Inc. utilizing their WHOOP strap to assess respiratory rate using Resting Heart Rate (RHR). Using a mobile application, WHOOP strap data is sent to the WHOOP system [85]. Their technique detected 20% of COVID-19 positive people two days before the beginning of symptoms and 80% of COVID-19 positive persons by the third day of symptoms. Philips has also created temporary patches for the identification of COVID-19 patients in their early stages [86] and disposable biosensors for the early identification of deterioration in COVID-19 patients [87]. It measures and sends a number of indicators of deterioration, including respiratory rate, heartbeat, activity level, body position, and ambulation, among other things, to the doctor.

Artificial Intelligence (AI) can be used to accurately differentiate one disease from another [88,89]. Recently, AI was used by researchers from the Institute of Technology and Harvard University to determine whether COVID-19 subjects could be accurately differentiated from only a forced-cough cell phone recording [86]. Their findings, which were based on cough recordings from over 5000 subjects, show their model performed accurately. COVID-19 subjects were officially tested 97.1 percent of the time, and 100% of them were found to be asymptomatic when they were tested. Cough recordings have been used in the past to accurately diagnose circumstances such as pneumonia and asthma. This shows how useful it is to put these solutions into wearable devices so that they can provide a non-invasive, realistic solution for diagnosis of diseases, pre-screening, and outbreak monitoring [87,90].

A study by the Scrips Research Translations Institute is called Digital Engagement Tracking for Early Control Treatment (DETECT). DETECT gathered data from smartwatches and activity trackers that partners agreed to use, as well as self-reported symptoms and test results [91]. DETECT recently reported [92] that data from wearable devices can be used to identify COVID-19 cases with more accuracy than just symptoms alone. There have been several other studies such as this one [63,93,94]. This has accelerated deployment to allow interested individuals to voluntarily share their sensor and clinical data in order to combat COVID-19.

As a result of COVID-19, it has become increasingly critical for healthcare to be accessible, work on a low threshold, and be rapid in monitoring, testing and diagnosing [95]. Suddenly, we are faced with a brand-new set of issues. To begin, ubiquity implies that IoT-based medical services must be freely available and accessible to a greater population. However, the majority of people lack access to professional medical equipment, and must use less expensive technology such as smartphones and smart- watches, which have restricted healthcare capabilities. Second, since the threshold is low, the cost and difficulties of adopting IoT-based medical services must be reduced. On the other hand, medical care procedures are often complicated and expensive. Additionally, many individuals struggle to embrace and use new technology such as IoT solutions for healthcare. Third, it is necessary to address the energy needs and computational effectiveness of medical sensors in order to provide continuous tracking and high-quality testing/diagnosis [95].

## 8. Regulations and Procedures for IoT during Pandemic

Previously, IoT adoption has been considerably lower in healthcare as a result of regulatory laws controlling privacy, data security, and approval processes. As a result of COVID- 19, emergency rules are being applied via established processes, and several innovative technologies have acquired emergency approvals driving IoT adoption. For instance, the United States Food and Drug Administration (USFDA, or FDA) has issued an Emergency Use Authorization for the electrocardiogram (ECG) low-ejection-fraction instrument created by Eko.

Eko is a company that makes digital health tools. They are based in the US and their tools assist health and clinicians providers in assessing cardiac complications associated with COVID-19 [96,97]. Eko makes it simpler for healthcare practitioners during ECG recordings to save, analyze, and communicate patient heart and lung sounds. Eko’s machine learning algorithms were carefully developed utilizing different real-world data sets and are clinically proven to aid physicians in detecting early indicators of cardiac disease. The World Health Organization (WHO) has warned people about the risk of becoming sick in extremely crowded hospitals and emergency rooms. As a result of the regulation, the use of telehealth and home care has increased to cut down on hospital and clinic visits. Therefore, several countries are encouraging the use of telehealth services and have decided to add many types of medical services to their public health programs that can be accessed through telehealth. When people start using telehealth, they are using more IoT applications and other technologies. This is because the technology is becoming more common in the healthcare field. Ref. [98] shows how IoT technologies, smart telemedicine diagnosis systems, and virtual care work together for people of different ages and backgrounds [99,100].

## 9. Discussion

The whole issue of IoT technology in healthcare is gaining interest from companies and academics, since it provides a novel method of communicating with healthcare professionals and patients alike. Furthermore, it is a promising instrument to aid the healthcare industry [13]. The purpose of the study is to summarize the literature on factors that might support or prevent health professionals from using IoT technology in their job. Indeed, many nations own IoT devices and employ them in the healthcare sector. Nevertheless, this does not guarantee that the professionals adopt and accept it completely [15,101]. Therefore, it becomes important to study and identify the factors that may facilitate or impede healthcare professionals’ use of IoT technologies. The review’s primary results indicate that a variety of effective factors influence IoT adoption at the individual, technology, security, health, and environmental levels. The usefulness and ease of use of the technology were identified as two of the most critical factors influencing IoT adoption in the research. Furthermore, those two factors considered TAM theory factors. Moreover also, it is the most frequent theory in these studies for studying healthcare professional acceptance and adoption of IoT technology according to the literature. Moreover, most studies in this review are based on quantitative methods. The majority of the studies were conducted in developing countries [14,49,53]. In addition, our findings show that healthcare professionals think social influence factors in individual factors, privacy risk issues in security factors, perceived severity and perceived health risk in health factors, and cost issues in environment factors, could limit the adoption of the IoT technology. As mentioned, it has been noted in the literature that IoT technology may play a role in empowering patients and healthcare professionals [47,58]. In fact, medical experts feel that IoT technology in healthcare supports and improves doctor-patient relationships [49,58]. Additionally, the results show that healthcare providers agree that IoT in healthcare could improve patient care. We also assessed the results regarding prior studies conducted. We found that all studies considered the privacy risk issue in security factors as barriers [10,47,53]. Our findings in this review shows the same result. Additionally, some studies mentioned factors such as perceived severity and perceived health risk as having no significant effect on the adoption of IoT in healthcare [43]. Notwithstanding, other studies mentioned them as having an indirect effect on adoption [10]. Our analysis in this review found them to be regarded as barriers. Moreover, some studies consider individual factors to have no significant role in the adoption of technology [40], but the most studies are consider these to be facilitators [40,45]. Similarly, this review found the individual factors to be facilitators. Environmental factors in prior studies have been classified as barriers to adoption [55], and technology factors in all studies are considered facilitators [40,43]. This relationship is highlighted by our findings.

Finally, the results of this systematic review provide a baseline, allowing for a more comprehensive understanding of the challenges and opportunities associated with healthcare professionals’ utilization of IoT technology.

### 9.1. Gaps and Implications for Future Research

The majority of the studies included in the review consists of physicians as professionals in healthcare. In contrast, there are no studies conducted on nurses as professionals in healthcare even though nursing is considered as an essential segment for caring for patients in the healthcare sector. Moreover, nurses are considered the primary key potential users of IoT technology in healthcare. They play a large role in the adoption of IoT technology. Furthermore, the present studies used quantitative methods more than qualitative methods and mixed methods. Indeed, using different design methods provide methodological flexibility.

### 9.2. Limitations

This study is not without limitations, which presents opportunities for further research. While this study provides an extensive compilation of current research about the factors influencing healthcare professional IoT adoption, it does have certain limitations. At the beginning, we searched the literature using just nine bibliographic databases, and we thoroughly evaluated the references of included studies as well. In addition, we include the papers citing those articles in order to reveal other possibly relevant publications. Secondly, in this study, we used a mixed-method systematic review approach to do a thorough review. It would have been better if we had used other methods, such as meta-narrative or realism review, to acquire a more complete picture of how IoT is used in healthcare. People who do research in this area will find it very useful. Third, this review only looked at data from studies that have been published. There was no extra contact with the authors to obtain more information or to make sure our classification was correct. Finally, we used a basic framework of adoption factors as the conceptual framework for categorizing elements identified as factors that affect IoT adoption in the studies included in this review and their repetitions in published studies. Even though we believe the framework is thorough and well-suited to introduce the adoption of IoT perceived by medical professionals, it is based on comprehensive, theoretical, and empirical research, but it may need further research.

However, future study may examine IoT adoption from a theoretical and empirical perspective.

### 9.3. Challenges of Effective IoT’s Adoption and Research Directions

COVID-19 has provided both possibilities and barriers to IoT adoption in the real world. Macroeconomically, IoT adoption must take into consideration the enormous cultural and economic shifts brought by COVID-19. Individual, community, and organizational behaviors have changed dramatically since the start of the global pandemic [102]. Prior to the pandemic, the worldwide IoT industry was developing. However, the accessibility and cost of device installation, and the security of data, have worsened. Microeconomically, IoT technology demands more quicker and revolutionary innovation in order to secure society’s functioning, encourage civic building, and react to any future crises. Different IoT industries are experiencing new issues, and finding solutions that work will be crucial in speeding up IoT growth and acceptance in these areas [103].

#### 9.3.1. Pervasive Challenges across All Verticals (A) Financial Challenges

Numerous firms have cut or eliminated investment in a variety of new or planned projects, including IoT projects. Another financial concern exacerbated by COVID- 19 is the increased workers’ cost for appliance installation in environments with social interaction limitations.

It is becoming more critical to design affordable, easy-to-install, and maintain IoT sensors and devices. For instance, it is vital to build low-power gadgets in order to save money on things such as battery replacement [104]. The Ultra Low Power SoC made by Dialog Semiconductor is a good example. It is used in battery-powered IoT devices [63], and has a battery life of more than a year in various IoT applications. Additionally, it is necessary to produce inexpensive plug-and-play sensing devices [92,105,106], as well as intelligent human-computer interaction [107], to simplify the installation and use of IoT systems.

#### 9.3.2. Data Protection and Privacy

During the COVID-19 pandemic, numerous countries have implemented a variety of emergency measures, including limits on mobility, social distance standards, and less rigorous privacy standards [108]. Following the pandemic’s conclusion, such legislation will need to be thoroughly evaluated to guarantee that individual rights and privacy are respected [63]. Researchers need to build strong data access regulations, security standards, and privacy-preserving systems for tracking, monitoring, and analyzing, among other things, to be ready for new epidemics and emergencies [109]. Indeed, individual data pools on personal devices might be seen as a viable replacement for conventional data centers. They do not actively collect and upload data, but rather transmit it to consumer devices through produced data. It is possible for people to share data for data analysis and decision-making with people they trust or with third-party apps. Personal gadgets also need reliable encryption and network communication technologies. The system should smoothly incorporate 5G, edge computing, and blockchain [94,110].

#### 9.3.3. Healthcare in COVID-19

COVID-19 is a healthcare catastrophe with obvious and immediate effects. According to a Juniper Research report, IoT platform revenue is predicted to expand by roughly 20% in 2020, from USD 55 billion in 2019 to USD 66 billion in 2020, and up from USD 35 billion in 2021. It has also found that revenues generated by IoT will exceed USD 4.3 billion by 2022–2023 [111], and according to [112] the annual economic impact of IoT in 2025 would be in the range of USD 2.7 to USD 6.2 trillion. Currently, three primary topics are driving IoT adoption in healthcare [111]

## 10. Conclusions

IoT has emerged as a new paradigm for improving healthcare. The health industry could potentially realize the advantages of IoT technologies as a result of the digital and information revolution [113,114]. The main benefits of the IoT are providing sustainable healthcare services, well-being, and more cost-effective treatment. The main goal of this study was to compile available literature related to IoT adoption and application for smart healthcare. This study examined, gathered, analyzed, and synthesized the essential data in a systematic manner. According to the results, the most commonly employed adoption theories for IoT adoption are TAM, TPB, TRA, and UTAUT. Furthermore, at the individual level, the key recognized adoption elements of IoT application in healthcare are social influence, attitude, and personal inattentiveness. Perceived usefulness, perceived ease of use, performance expectancy, and effort expectations are the IoT adoption factors at the technological level. Furthermore, perceived privacy risk is the most important issue at the security level. At the health level, the primary factors are perceived severity and perceived health risk, respectively; at the environmental level, the main factors are financial cost and facilitating conditions. The majority of the respondents participating in the included papers were physicians, patients, and healthcare professionals. Existing studies include various sorts of IoT applications such as wearable devices, monitoring devices, rehabilitation devices, telehealth and behavior modification, smart cities, and smart homes. In the year 2020, the majority of research on IoT adoption was done in France and Pakistan. This systematic review analyzes the critical characteristics that allow healthcare practitioners to recognize the constraints and opportunities for implementing IoT applications. Finally, we assess the anticipated impact of COVID-19 on IoT adoption in healthcare. After we thoroughly examined many of the studies in the literature, reports from prominent consulting companies, and interviews with specialists from a variety of sectors, we recognized the potential impact of COVID-19 on IoT adoption in various sectors, including healthcare, transportation, industrial IoT, smart homes, smart buildings, and smart cities, and how it pushed technology adoption and innovation. In addition, we address the different efforts that have been launched in a variety of areas in the aftermath of the epidemic. Furthermore, we discussed numerous obstacles that must be overcome as well as critical research paths that must be emphasized in order to expedite IoT adoption across the healthcare sector and other industries. Based on the review results, we can conclude that the main challenges and research directions for facilitating IoT adoption in various sectors are more related to wearables that are low-energy or harvest energy, wearables that are research-grade, AI algorithms for healthcare devices (for instance Artificial intelligence (AI)), and more accessible healthcare services. Furthermore, reducing the cost of developing, installing, and using IoT solutions and systems, as well as data security and privacy remains as concern in healthcare and other related industries. In general, adoption of the IoT is still limited to a few application areas. The results of this systematic study provide a common base for discussing the problems and opportunities associated with the adoption and usage of IoT by healthcare providers. Additionally, this study evaluated the work that has been done in this field, whether it be models and frameworks offered for enhancing or adopting IoT in healthcare, or proposed solutions to enable the realization of in real-world settings. However, capturing the full benefits of new technology while achieving a sustainable socioecological transition, still remains a challenge for government and the healthcare industry.

## Figures and Tables

**Figure 1 sensors-22-05377-f001:**
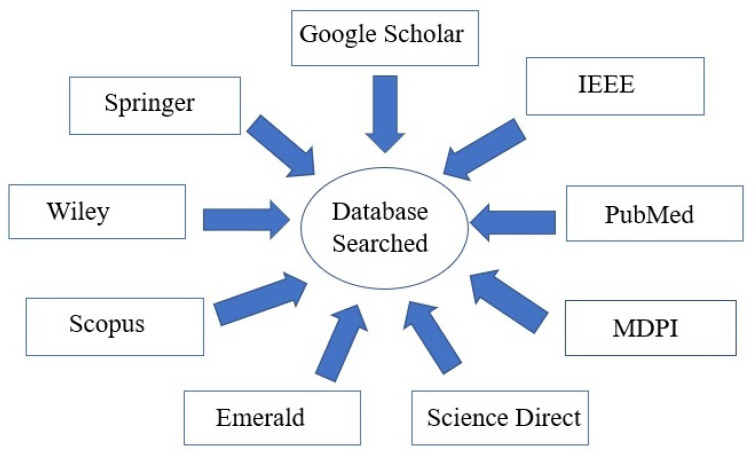
Nine Digital Data Bases.

**Figure 2 sensors-22-05377-f002:**
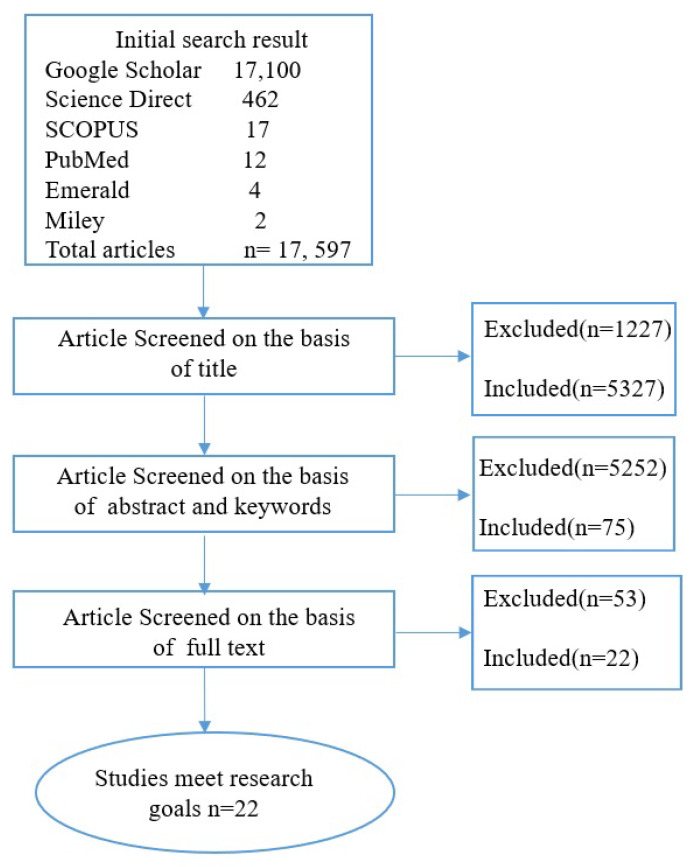
Research Process.

**Figure 3 sensors-22-05377-f003:**
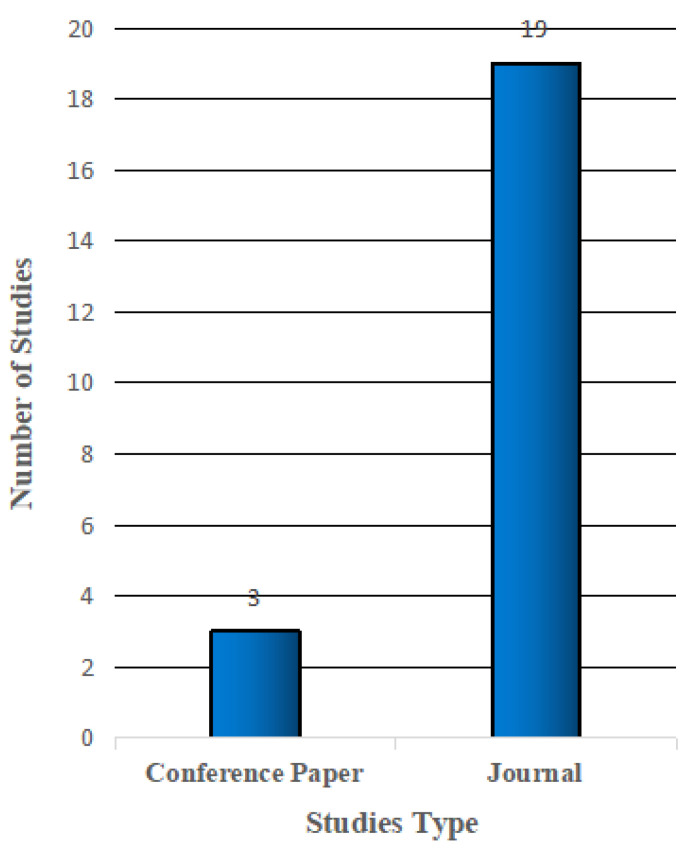
Studies Type Included in Review.

**Figure 4 sensors-22-05377-f004:**
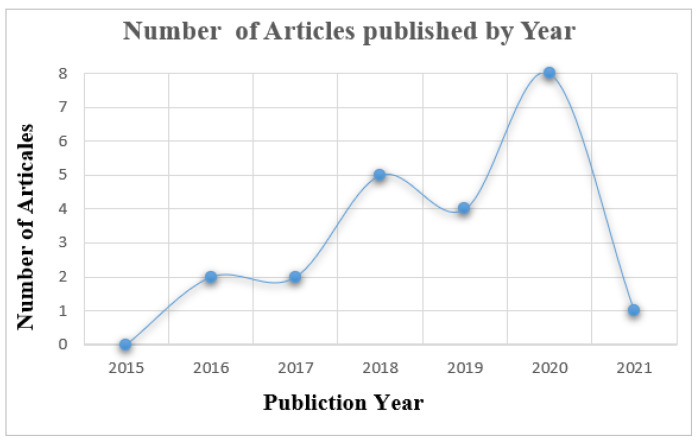
Number of Articles Published by Year.

**Figure 5 sensors-22-05377-f005:**
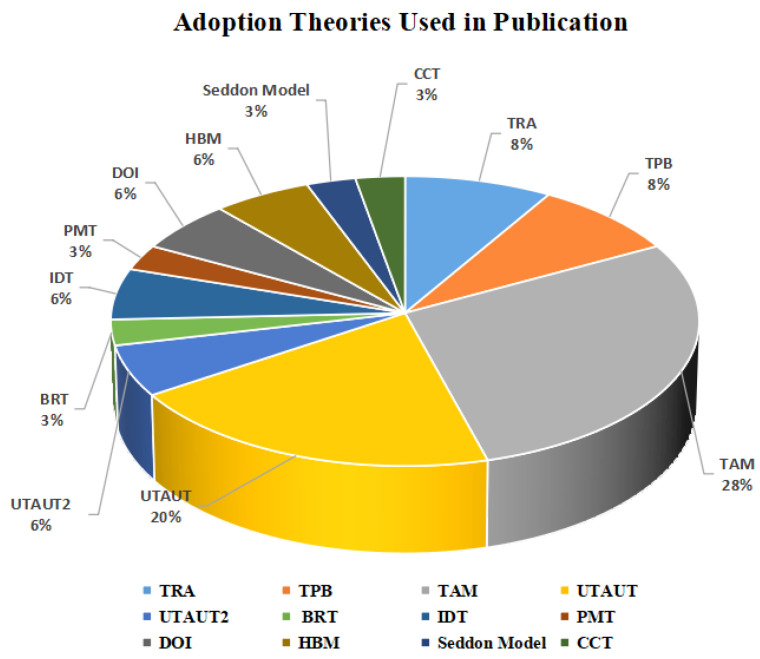
Adoption theories used in publication.

**Figure 6 sensors-22-05377-f006:**
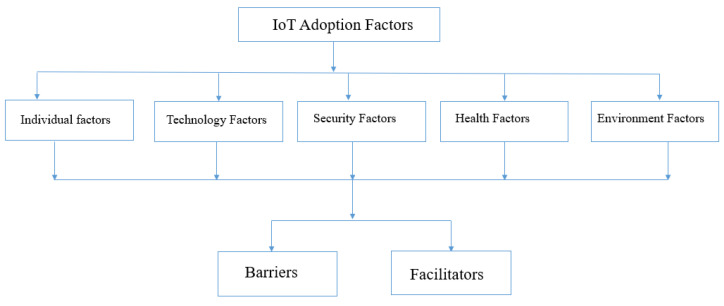
Factors Characteristics.

**Figure 7 sensors-22-05377-f007:**
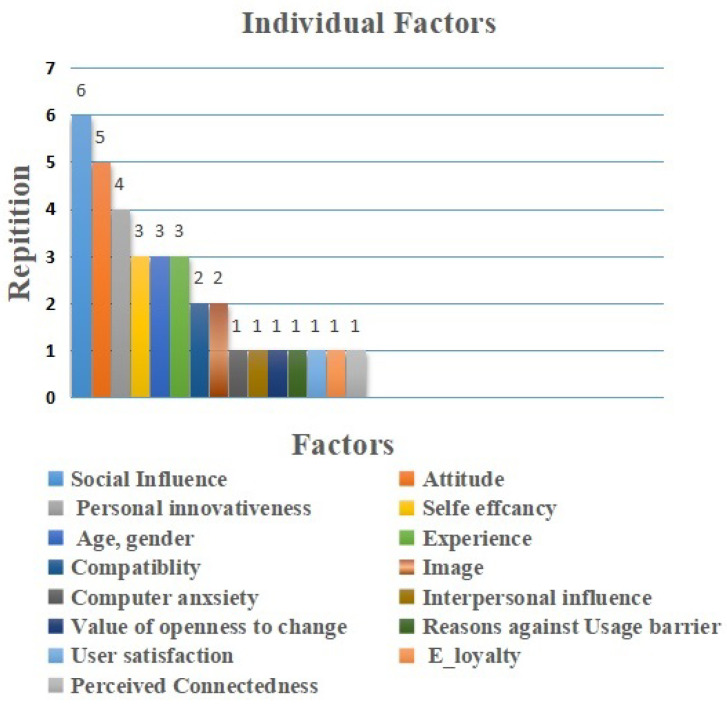
Individual Factors.

**Figure 8 sensors-22-05377-f008:**
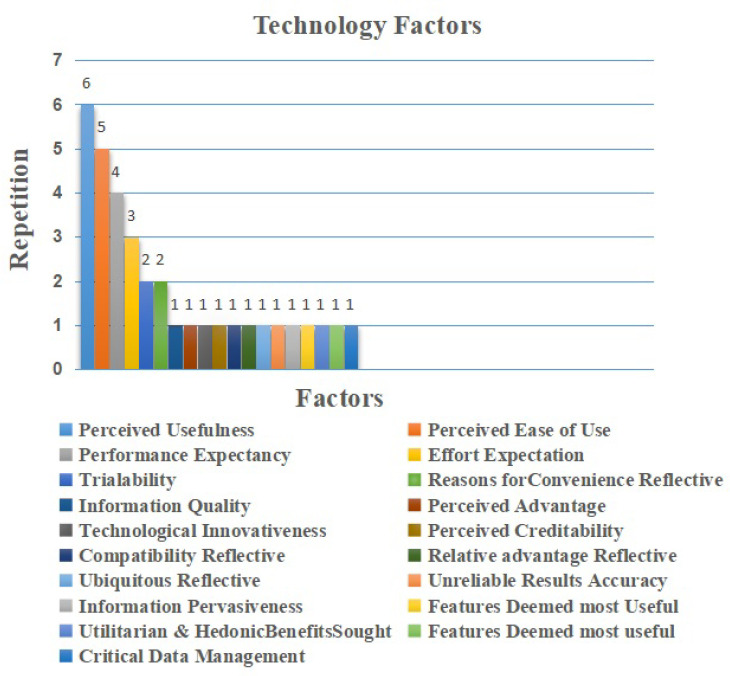
Technology Factors.

**Figure 9 sensors-22-05377-f009:**
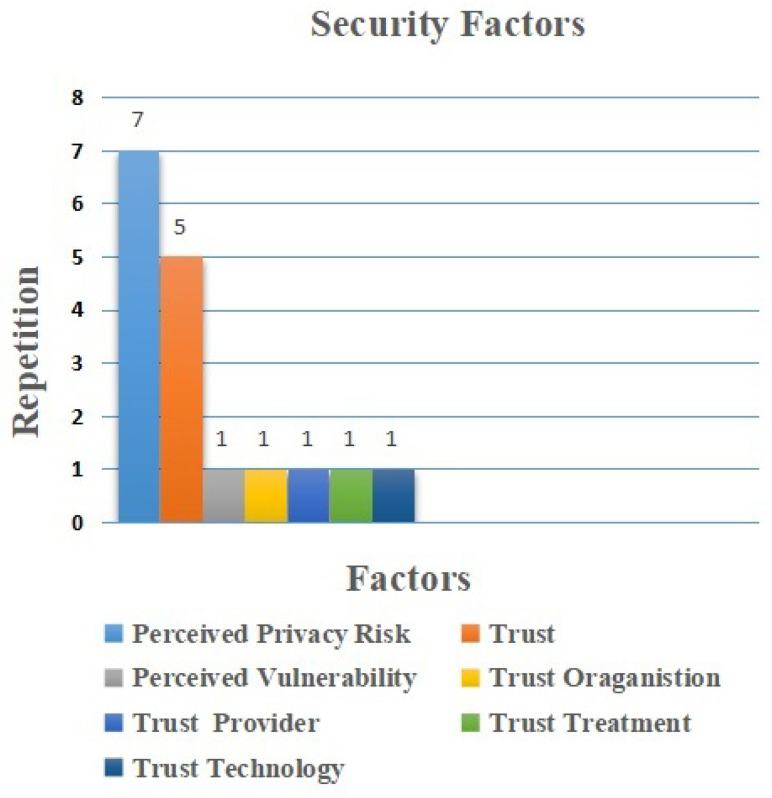
Security Factors.

**Figure 10 sensors-22-05377-f010:**
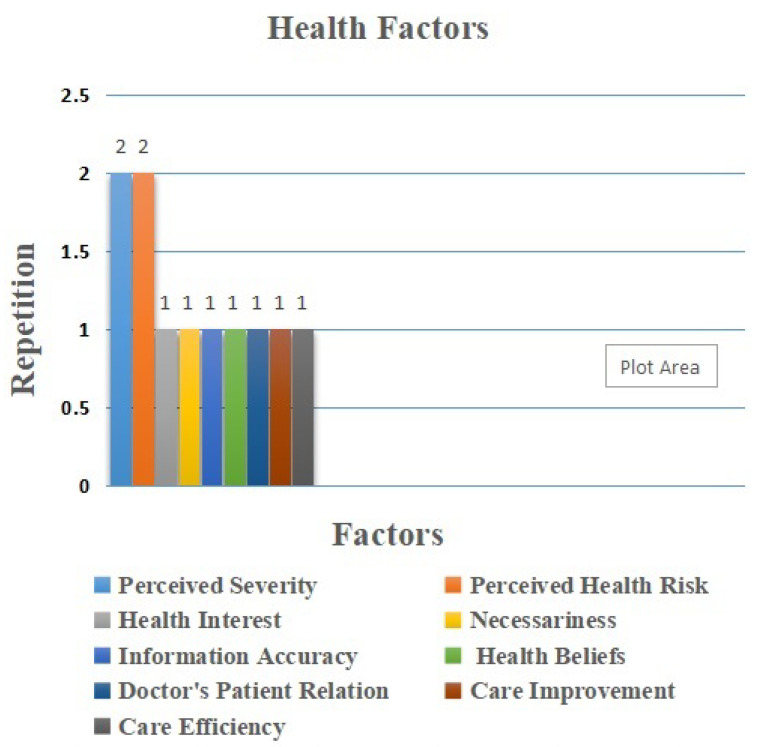
Health Factors.

**Figure 11 sensors-22-05377-f011:**
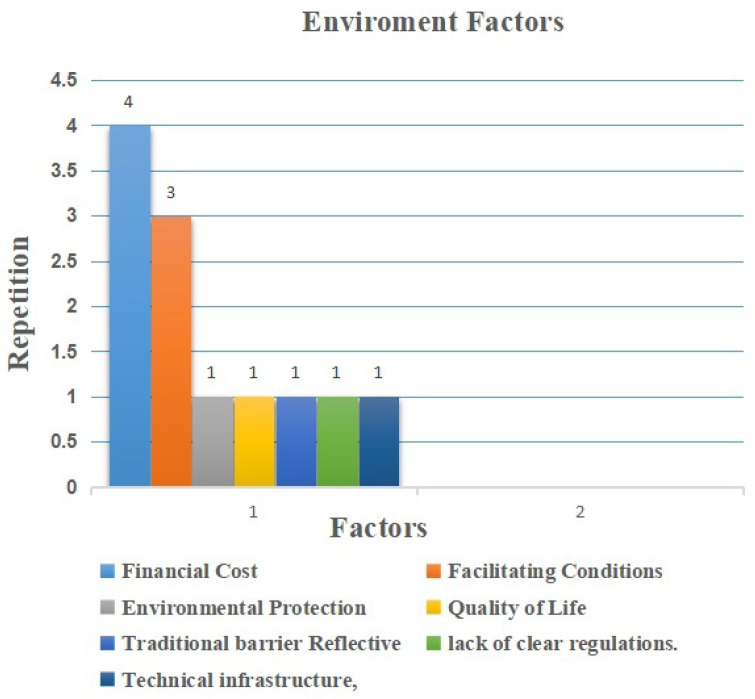
Environment Factors.

**Figure 12 sensors-22-05377-f012:**
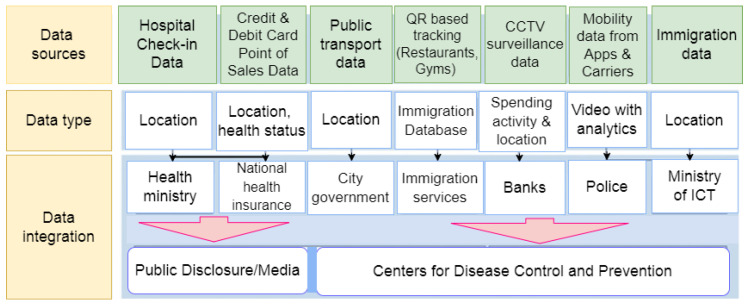
An Architecture of a Test and Trace IoT.

**Table 1 sensors-22-05377-t001:** Quality assessment scores for included review papers.

PID	Q1	Q2	Q3	Q4	Q5	Scores
P1	1	1	1	0.5	1	4.5
P2	1	1	1	1	1	5
P3	1	1	1	1	1	5
P4	1	1	1	1	1	5
P5	1	1	1	1	1	5
p6	1	1	1	1	1	4
P7	1	1	0	0.5	1	3.5
P8	1	1	1	1	1	5
P9	1	1	1	1	1	5
P10	1	0	0	0	0	1
P11	1	1	1	1	1	5
P12	1	1	1	1	1	5
P13	1	1	1	1	1	5
P14	1	1	0	1	1	4
P15	1	1	0.5	1	1	4.5
P16	1	1	0	0	0	2
P17	1	1	1	1	1	5
P18	1	1	1	1	1	5
P19	1	1	1	1	1	5
P20	1	0	0	0	0	1
P21	1	1	1	1	1	5
P22	1	1	1	1	1	5

**Table 2 sensors-22-05377-t002:** Elements of the data extraction form with descriptions.

Data Extraction	Description
Study ID	A unique identifier
Study Title	Title of each identified during the search.
Author(s)	Author name.
Year	Year of publication.
Type of Participants	The type of user the paper conduct them.
Research Design	Identification of the research methodology.
Studies Place	Country/region where the research was undertaken.
Theoretical Frameworks	Theory/model used by the selected papers.
Adoption’s Theory	Type of theory adoption used in the studies.
Constructs	The constructs/factors used in the frameworks.
Data collection strategy	Approach used to collect the data.
Sample	Research participants.
Type analysis and software	Software and type of analysis in the papers to obtain the result.
Degree of article	A number indicating how much this study met the criteria for research quality.

**Table 4 sensors-22-05377-t004:** Data extraction form.

SID	Study	Year	Type of Participants	Research Design	Studies Place	Theoretical Frameworks	Data Collection	Sample	Analysis and Software
S1	[44]	2017	Respondents in India	Not Mentioned	India	Yes	Survey	314	Partial Least Square SEM
S2	[40]	2020	Users of IoT-based healthcare devices	Quantitative Method	France	Yes	Survey	268	PLS-SEM
S3	[45]	2020	Younger physicians	Quantitative Method	Srilankan	Yes	Survey	375	SPSS
S4	[46]	2021	Patients	Quantitative Method	France	Yes	Online Survey	267	Partial Least Approach—Structural Equation Modeling
S5	[47]	2018	Older adults	Quantitative Method	Indian	Yes	Survey	815	PLS-SEM
S6	[10]	2018	End user IoT Product	Quantitative Method	Not Mentioned	Yes	Online Survey	426	SEM-PLS, and XLSTAT-PLSPM
S7	[48]	2020	The public user	Qualitative Method	Malaysia	No	Survey	Not Mentioned	Not Mentioned
S8	[49]	2020	Clinicians	Qualitative Method	Pakistan	Yes	Questionnaire	Over 479	PLS SEM
S9	[14]	2020	Professionals or service administrators in healthcare	Mix Method	Saudi Arabia	Yes	Semi-Structured Interviews and Survey Data	Not Mentioned	NVIVO Software
S10	[50]	2018	applications	Not Mentioned	Not Mentioned	Yes	Not Mentioned	Not Mentioned	Fuzzy Logic
S11	[51]	2020	Patients	Quantitative Method	Not Mentioned	Yes	Questionnaire	117	PLS SEM
S12	[52]	2020	Device users	Quantitative Method	Germany and Sweden	Yes	Questionnaire	97	PLS SEM
S13	[53]	2020	Doctors	Quantitative Method	Iraq	Yes	Online Survey	250	SPSS
S14	[54]	2016	Physicians	Mixed-Methods	Israel	No	Questionnaire, Personal, and semi- Structured Interviews.	176	Microsoft Excel, and SPSS
S15	[42]	2019	Cardiologist Diabetologist Nutritionist	Quantitative Method	Not Mentioned	Yes	Online Survey	221	SEM
S16	[55]	2016	User wearable	Focus Group	Not Mentioned	No	Not Mentioned	Not Mentioned	Not Mentioned
S17	[43]	2017	Medical Doctors, Nursing Staff, and Patients	Quantitative Method	Pakistan	Yes	Survey	100	SPSS23
S18	[56]	2019	Users	Quantitative Method	Omani	Yes	Questionnaires	387	SPSS 25 and AMOS 25 statistics
S19	[57]	2019	Patient	Quantitative Method	Kingdom of Saudi Arabia	Yes	Survey	407	SEM
S20	[58]	2018	Patient	Quantitative Method	Latin-America,	No	Not Mentioned	Not Mentioned	Not Mentioned
S21	[59]	2018	Medical staff Care Services, Medical specialties, Covered Medical Facilities	Quantitative Method	Spain	Yes	Questionnaire	256	SPSS MEDIATE
S22	[60]	2019	Customers of Wearable Technology	Quantitative Method	Hong Kong	Yes	Online Survey	171	SmartPLS v3.28

## Data Availability

This article contains no data or material other than the articles used for the review and referenced.

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
