# Peer review of "IoT Adoption and Application for Smart Healthcare: A Systematic Review"

_sensors, 2022, doi:10.3390/s22145377_

Round 1

Reviewer 1 Report

The paper is interesting but the organization needs to improve as there are many sections in the paper such as methodology should include information sources, selection criteria, and data extraction method, etc. In addition, the Discussion section should include the sub-section of challenges of effective IoT's adoption and research directions, gaps and implications for future research, etc.. Last but not least, the Conclusion section should include major findings and limitations of this study.

Reviewer 2 Report

The purpose of this study is to accumulate existing knowledge about the factors that influence medical professionals to adopt IoT applications in the healthcare sector.Several recent studies have offered important insights into IoT adoption in healthcare. This paper just reviews the effective factors of IoT adoption in a systematic manner. This topic is not original, but relevant in the field. there are some fatal errors in the drafting of the manuscript. Also, the organization of the sections must be revised. the references Are appropriate.I have other questions and comments:

1. Section 12. Limitations : The text from line 464 to 487 is duplicated from line 488 to 507. Please check that the final version of the paper is submitted.

2. English is generally very good, but needs to be polished further.

3. Line 684 : "Table ??", please check all table number.

4. Text in line 669 is duplicated in line 686

5. Sections 13, 14, 15, 16 are not consistent, please revise the plan.

Concluding, it seems that this version is not well finalized and requires more revision.

Round 2

Reviewer 2 Report

The authors reacted properly to my pointed issues.